# MultiPull: Detailing Signed Distance Functions by Pulling Multi-Level Queries at Multi-Step

**Takeshi Noda**[1*]     **Chao Chen**[1*]     **Weiqi Zhang**[1]     **Xinhai Liu**[2]
**Yu-Shen Liu**[1†]     **Zhizhong Han**[3]

[1]School of Software, Tsinghua University, Beijing, China
[2]Tencent, Hunyuan, Beijing, China
[3]Department of Computer Science, Wayne State University, Detroit, USA
yeth21@mails.tsinghua.edu.cn   chenchao19@tsinghua.org.cn
zwq23@mails.tsinghua.edu.cn   adlerxhliu@tencent.com
liuyushen@tsinghua.edu.cn   h312h@wayne.edu

## Abstract

Reconstructing a continuous surface from a raw 3D point cloud is a challenging task. Recent methods usually train neural networks to overfit on single point clouds to infer signed distance functions (SDFs). However, neural networks tend to smooth local details due to the lack of ground truth signed distances or normals, which limits the performance of overfitting-based methods in reconstruction tasks. To resolve this issue, we propose a novel method, named MultiPull, to learn multi-scale implicit fields from raw point clouds by optimizing accurate SDFs from coarse to fine. We achieve this by mapping 3D query points into a set of frequency features, which makes it possible to leverage multi-level features during optimization. Meanwhile, we introduce optimization constraints from the perspective of spatial distance and normal consistency, which play a key role in point cloud reconstruction based on multi-scale optimization strategies. Our experiments on widely used object and scene benchmarks demonstrate that our method outperforms the state-of-the-art methods in surface reconstruction. Project page: https://takeshie.github.io/MultiPull

## 1   Introduction

Reconstructing surfaces from 3D point clouds is an important task in computer vision. It is widely used in various real-world scenarios such as autonomous driving, 3D scanning and other downstream applications. Recently, using neural networks to learn signed distance functions from 3D point clouds has made huge progress [1, 2, 3, 4, 5, 6, 7, 8]. An SDF represents a surface as the zero-level set of a 3D continuous field, and the surface can be further extracted using the marching cubes algorithm [9]. In supervised methods [10, 11, 12, 13], a continuous field is learned using signed distance supervision. Some methods employ multi-level representations [14, 15], such as Fourier layers and level of detail (LOD) [16, 17], to learn detailed geometry. However, these methods require 3D supervision, including ground truth signed distances or point normals, calculated on a watertight manifold. To address this issue, several unsupervised methods [18, 19, 20, 21, 22] were proposed to directly infer an SDF by overfitting neural networks on a single point cloud without requiring ground truth signed distances and point normals. They usually need various strategies, such as geometric constraints [18, 19, 20] and consistency constraints [22, 23], for smoother and more completed signed distance field. However, the raw point cloud is a highly discrete approximation of the surface, learning

---

[*]Equal contribution. [†]The corresponding author is Yu-Shen Liu.

38th Conference on Neural Information Processing Systems (NeurIPS 2024).

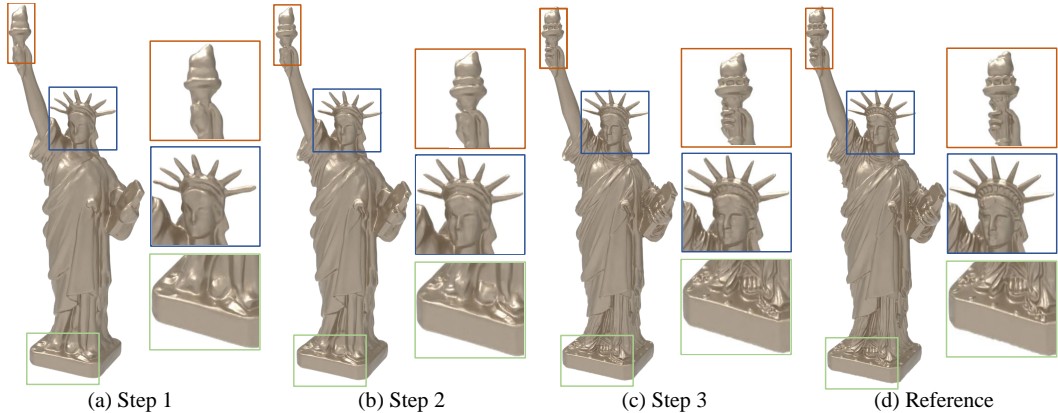

| (a) Step 1 | (b) Step 2 | (c) Step 3 | (d) Reference |

Figure 1: Visualization of the 3D shape reconstruction. In (a), (b) and (c), SDFs are learned from a point cloud by optimizing multi-level query points at multi-step. At each step, we optimize query points at one level with frequency features at this specific level as conditions. This enables the network to progressively recover coarse-to-fine geometry details.

SDFs directly from the point cloud is often inaccurate and highly ambiguous. This makes it hard for the network to learn accurate SDFs on local details, resulting in over-smooth reconstruction.

To address this issue, we propose *MultiPull*, to learn an accurate SDF with multi-scale frequency features. It enables network to predict SDF from coarse to fine, significantly enhancing the accuracy of the predictions. Furthermore, to optimize the SDF at different scales simultaneously, we introduce constraints on the pulling process. Specifically, given query points sampled around 3D space as input, we use a Fourier transform network to represent them as a set of Fourier features. Next, we design a network that can leverage multi-scale Fourier features to learn an SDF fields from coarse to fine. To optimize the signed distance fields with multi-scale features, we introduce a loss function based on gradient consistency and distance awareness. Compared with Level of Detail (LOD) methods [16, 17, 24], we can optimize the signed distance fields effectively without a need of signed distance supervision, recovering more accurate geometric details. Evaluations on widely used benchmarks show that our method outperforms the state-of-the-art methods. Our contribution can be summarized as follows.

- We propose a novel framework that can directly learn SDFs with details from raw point clouds, progressing from coarse to fine. This provides a new perspective for recovering 3D geometry details.
- We introduce a multi-level loss function based on gradient consistency and distance awareness, enabling the network to geometry details.
- Our method outperforms state-of-the-art methods in surface reconstruction in terms of accuracy under widely used benchmarks.

## 2   Related Work

Classic methods for geometric modeling [25, 26, 27, 28] have attempted to analyze the geometric modeling of objects, which do not require large-scale datasets. With the advent of extensive and intricate 3D datasets like ShapeNet [29] and ABC [30], learning-based methods have achieved significant advancements [18, 12, 31, 32, 22, 33, 34, 35, 36, 37, 38]. These approaches learn implicit representations from various inputs, including multi-view images[39, 40, 41, 42, 43, 44], point clouds [45, 46, 47], and voxels [48, 49, 50].

**Learning Implicit Functions with Supervision**. Supervised methods have made significant progress in recent years. These methods leverage deep learning networks to learn priors from datasets or use real data for supervision [10, 11, 51, 52, 53, 54] to improve surface reconstruction performance. Some supervised approaches use signed distances and point normals as supervision, or leverage occupancy labels to guide the network's learning process. In order to improve the generalization ability of neural networks and learn more geometric details, some studies learn geometry prior of shapes through supervised learning.

**Learning Implicit Functions with Conditions**. To alleviate the dependence on supervised information, recent studies focus on unsupervised implicit reconstruction methods. These methods do not require pretrained priors during optimization. For example, NeuralPull (NP) [21] learns SDF by pulling query points in nearby space onto the underlying surface, which relies on the gradient field of the network. CAP [55] further complements this by forming a dense surface by additionally sampling dense query points. GridPull [23] generalizes this learning method to the grid, by pulling the query point using interpolated signed distances on the grid. In addition, some studies explore surface reconstruction more deeply and propose innovative methods, such as utilizing differentiable Poisson solutions [56], or learning signed [57, 58, 19, 51] or unsigned functions [59, 55] with priors. However, inferring implicit functions without 3D supervision requires a lengthy convergence process, which limits the performance of unsupervised methods on large-scale point cloud datasets. To address this, we propose a fitting-based frequency feature learning strategy that efficiently learns implicit fields without the need for additional supervision.

**Learning Implicit Functions with LOD**. Level-Of-Detail (LOD) models [16, 17, 24] are used to simplify code complexity and refine surface details through the architecture of multi-level outputs. Previous studies have explored multi-scale architectures in various reconstruction tasks. For example, NGLOD [16] uses octree-based feature volumes to represent implicit surfaces, which can adapt to shapes with multiple discrete levels of detail and enable continuous level-of-detail switching through SDF interpolation. MFLOD [17] applies Fourier layers to LOD, which can offer better feasibility in Fourier analysis. However, it is difficult to optimize multi-level features simultaneously to learn 3D shapes. To address this issue, we propose a novel strategy to optimize multi-level frequency features, allowing the network to progressively learn geometric details from coarse to fine.

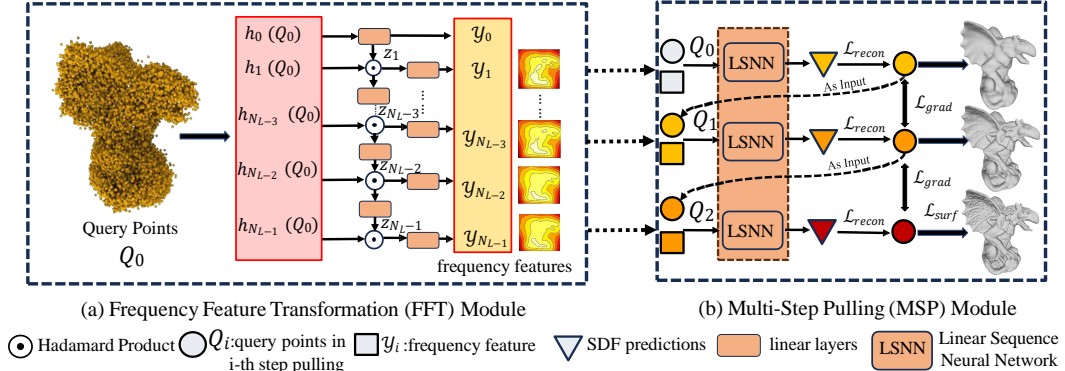

Figure 2: Overview of our method: (a) Frequency Feature Transformation (FFT) module and (b) Multi-Step Pulling (MSP) module. In (a), we learn Fourier bases $h_i(Q)$ from query points $Q$ using the Fourier layer and obtain multi-level frequency features $y_i$ through Hadamard product. In (b), using multi-level frequency features from (a) and a linear network **LSNN** with shared parameters, we calculate the distance(D) of $Q_i$ to its corresponding surface target point $Q_t$ to predict a more accurate surface. We visualize the predicted SDF distribution map corresponding to the frequency features in (a) and the reconstruction from each step of SDF predictions on the right side of (b).

## 3 Method

**Overview**. The overview of MultiPull is shown in Fig. 2. We design a neural network to learn an implicit function $f$ from a single 3D point cloud by progressively pulling a set of query points $Q_0$ onto the underlying surface, where $Q_0$ is randomly sampled around the raw point cloud $S$. Our network mainly consists of two parts as follows.

(1) The Frequency Feature Transformation (FFT) Module ( Fig. 2(a)) aims to convert the query points $Q_0$ into a set of multi-level frequency features $Y = \{y_i, i \in [0, N_L - 1]\}$. The key insight for introducing frequency features lies in a flexible control of the degree of details. (2) The Multi-Step Pulling (MSP) Module (Fig. 2(b)) is designed to predict $f$ with coarse-to-fine details under the guidance of frequency features $Y$. At the $i$-th step, we pull $Q_i$ to $Q_{i+1}$ using the predicted signed distances $s_i = f(Q_i, y_i)$ and the gradients at $Q_i$, according to its feature $y_i$. To this end, we constrain query points to be as close to their nearest neighbor point on $S$.

## 3.1 Frequency Feature Transformation (FFT) Module

We introduce a neural network to learn frequency features $Y$ from point clouds. The network manipulates input $Q_0$ through several linear layers to obtain an initial input $z_0$ and a set of Fourier basis $h_i(Q_0), i \in [0, N_L - 1]$, formulated as follows.

$$\begin{cases} h_i(Q_0) = \sin(\omega_i Q_0 + \phi_i), \\ \quad z_0 = h_0(Q_0), \end{cases} \tag{1}$$

where $\omega_i$ and $\phi_i$ are the parameters of the network, and $N_L$ is the number of layers of the network.

To effectively represent the expression of the raw input in the frequency space, we choose the sine function as the activation function and employ the Hadamard product to compute the intermediate frequency feature output. Since the Hadamard product allows the representation of frequency components as the product of two feature inputs, denoted as $a$ and $b$, which can be formulated as:

$$\sin(a)\sin(b) = \frac{1}{2}(\sin(a + b - \frac{\pi}{2}) + \sin(a - b + \frac{\pi}{2})). \tag{2}$$

Through Eq. (2), we can calculate the frequency component $z_i$ of $h_i(Q_0)$, and then obtain the output $y_i$ of the $i$-th layer, formulated as:

$$\begin{cases} z_i = h_i(Q_0) \odot (W_i z_{i-1} + b_i), i \in [1, N_L - 1], \\ y_i = W_i z_i + b_i, \end{cases} \tag{3}$$

where $\odot$ indicates the Hadamard product, $W_i, b_i$ are parameters of the network.

Frequency networks based on the Multiplication Filter Network (MFN) [14] typically employ uniform or fixed-weight initialization for network parameters in practice. This approach overlooks the issue of gradient vanishing in deep network layers during the training process, leading to underfitting and making the network overly sensitive to hyperparameter changes. To address this challenge, we propose a new initialization scheme that thoroughly considers the impact of network propagation, aiming at ensuring a uniform distribution of initial parameters. Specifically, we dynamically adjust initial weights, which can be formulated as:

$$\Psi_i = \sqrt{\eta \times \sin(i\pi/N_L)}, i \in [1, N_L - 1], \tag{4}$$

where $N_L$ and $\eta$ are the number of layers and the parameters of the network, respectively. We leverage the standard deviation as the initialization range to ensure that the parameters in Eq. (3) are within a reasonable range. As shown in Fig. 3, we compared the parameter distributions of different linear layers. The initialization scheme based on MFN results in gradient vanishing and small activations in deeper linear layers. In contrast, our initialization scheme ensures that the parameters of each linear layer follow a standard normal distribution.

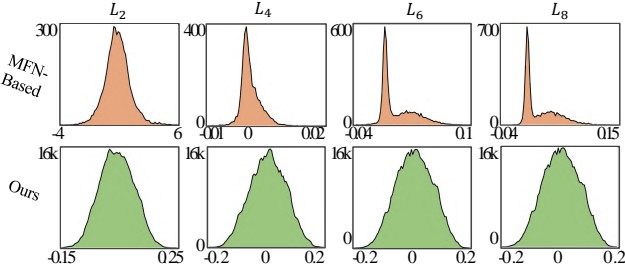

Figure 3: Comparison of parameter distributions of different linear layers especially in $(L_2, L_4, L_6, L_8)$. We show the different initialization strategies on the results of the reconstruction task and the visualization effects in Appendix B.

## 3.2 Multi-Step Pulling (MSP) Module

In Fig. 2(b), we demonstrate our idea of learning an accurate implicit function $f$ with multiple frequency features. Given a set of frequency feature $Y$, we use frequency features $y_i$ in $Y$ as the

input along with query points $Q_i$ for the MSP module. We follow NP[21] to construct initial query points and calculate the stride and direction of $Q_i$ at $i$-th step for pulling it to the target surface point. Furthermore, we use the direction of the gradient as $\nabla f(Q_i, y_i)$ and signed distance $f(Q_i, y_i)$ for the pulling, where $\nabla f(Q_i, y_i)$ represents the fastest increase in signed distance in 3D space, pointing away from the surface direction. Therefore, $Q_i = Q_{i-1} - f(Q_{i-1}, y_{i-1}) \cdot \nabla f(Q_{i-1}, y_{i-1}) / \parallel \nabla f(Q_{i-1}, y_{i-1}) \parallel_2$. For each step of pulling the query points $Q_i$, it corresponds to a nearest point $q_i$ on the surface, and the distance between query points and surface points can be described as $D_i = ||Q_i - q_i||_2^2$. Based on this, we initiate the optimization by pulling query points $Q_i$ the target points $q_i$ progressively. Therefore, we can obtain the combined loss $\mathcal{L}_{\text{pull}}$ under optimal conditions:

$$\mathcal{L}_{\text{pull}} = \sum_{i=1}^{I} D_i, i \in [1, I], \tag{5}$$

where $I$ is the step of moving operation. However, optimizing all query points accurately through this equation alone is challenging when merely constraining surface points. This is because the query points $Q_i$ may be located across multiple spatial scales with inconsistent gradient directions, indicating that simultaneous optimization becomes challenging. Consequently, some outlier points may not be effectively optimized. Additionally, for sampling points near target points, some surface constraints are required to enable the network to accurately predict their corresponding zero level-set to avoid optimization errors. Therefore, we will further advance Eq. (5) from the perspectives of distance constraints, gradient consistency, surface constraints in Sec. 3.3 to enhance network performance.

### 3.3 Loss Function

**Distance-Aware Constraint**. Inspired by FOCAL [60], we design a novel constraint with distance-aware attention weights $\alpha$ to ensure that the network pays more attention to the optimization of underfitting query points in space and optimizes the SDFs simultaneously. This allows query points at different distances from the surface to be optimized properly, and assigns higher attention weights for outlier and underlying points:

$$\begin{cases} \alpha_1, \alpha_2 = softmax(D_1, D_2), \\ \mathcal{L}_{\text{recon}} = \alpha_1 D_1 + \alpha_2^{\gamma} D_2 + D_3, \end{cases} \tag{6}$$

where $\alpha_1$ and $\alpha_2$ are calculated from $D_1$, $D_2$ by the softmax activation, $\gamma$ is a scaling coefficient we set to 2 by default. Here, we only consider 3 steps, which is a trade-off between performance and efficiency.

**Consistent Gradients**. We additionally introduce consistency constraints in the gradient direction. This loss encourages neighboring level sets to keep parallel, which reduces the artifacts off the surface and smooths the surface. We add a cosine gradient consistency loss function to encourage the gradient direction at the query points to keep consistent with the gradient direction at its target point on the surface, which aims to improve the continuity of the gradient during the multi-step pulling. We use $Q_1, Q_2$ and $Q_3$ to represent the query points that have been continuously optimized by the multiple steps. We take the one with the lowest similarity score to measure the overall similarity.

$$\begin{cases} L_{\nabla}(Q_i) = \cos(\nabla f(Q_i, y_i), \nabla f(Q_0, y_0)), \\ \mathcal{L}_{\text{grad}} = 1 - \min\{L_{\nabla}(Q_1), L_{\nabla}(Q_2), L_{\nabla}(Q_3)\}, \end{cases} \tag{7}$$

where $L_{\nabla}(Q_i)$ represents the loss of cosine similarity between query points $Q$ and target surface points $q$.

**Surface Constraint**. We introduce the surface constraint for the implicit function $f$, aiming to assist the network in approaching the zero level-set on the surface at final step. Hence, we constrain the $f(Q_I, y_I)$ approaches zero at the final step:

$$\mathcal{L}_{\text{surf}} = \parallel f(Q_I, y_I) \parallel . \tag{8}$$

**Joint Loss Function**. Overall, we learn the SDFs by minimizing the following loss function $\mathcal{L}$.

$$\mathcal{L} = \mathcal{L}_{\text{recon}} + \beta \mathcal{L}_{\text{grad}} + \delta \mathcal{L}_{\text{surf}}, \tag{9}$$

where $\beta$ and $\delta$ are balance weights. In the subsequent ablation experiments, we validated the effectiveness of different loss functions.

# 4 Experiments

In this section, we evaluate the performance of MultiPull in surface reconstruction by conducting numerical and visual comparisons with state-of-the-art methods on both synthetic and real-scan datasets. Specifically, in Sec. 4.1, we experiment on synthetic shape datasets with diverse topological structures. Furthermore, in Sec. 4.2, we report our results across various scales on real large-scale scene datasets. Meanwhile, we consider FAMOUS as the verification dataset in the ablation studies to compare the effectiveness of each module in MultiPull in Sec. 4.3.

## 4.1 Surface Reconstruction for Shapes

**Datasets and Metrics**. For the single shape surface reconstruction task, we perform evaluation on multiple datasets including ShapeNet [29], FAMOUS [10], Surface Reconstruction Benchmark (SRB) [45] Thingi10K [61] and D-FAUST [62]. We conduct validation experiments on 8 subcategories within the ShapeNet dataset, while the remaining datasets are experimented on the complete dataset. For metric comparison, we leverage L1 and L2 Chamfer Distance $CD_{L1}$ and $CD_{L2}$, Normal Consistency (NC), and F-Score as evaluation metrics.

**ShapeNet**. We evaluate our approach on the ShapeNet[29] according to the experimental settings of GP [23] . We compared our methods with methods including ATLAS [63], DSDF [51], NP [21], PCP [64], GP [23], as shown in Tab. 1. We report $CD_{L2}$, NC and F-Score metrics for ShapeNet, where we randomly sample 10,000 points on the reconstructed object surface for evaluation. MultiPull outperforms the state-of-the-art methods. Compare to previous gradient-based methods in Fig. 4, our method performs better by revealing more local details of these complex structures. We provide detailed results in Appendix C.

Table 1: Reconstruction accuracy on ShapeNet in terms of $CD_{L2}$, NC and F-Score with thresholds of 0.002 and 0.004.

| Methods | $CD_{L2} \times 100$ | NC | F-Score$^{0.002}$ | F-Score$^{0.004}$ |
|---------|---------|-----|-----|-----|
| ATLAS | 1.368 | 0.695 | 0.062 | 0.158 |
| DSDF | 0.766 | 0.884 | 0.212 | 0.717 |
| NP | 0.038 | 0.939 | 0.961 | 0.976 |
| PCP | 0.0136 | 0.9590 | 0.9871 | 0.9899 |
| GP | 0.0086 | 0.9723 | 0.9896 | 0.9923 |
| Ours | **0.0075** | **0.9737** | **0.9906** | **0.9932** |

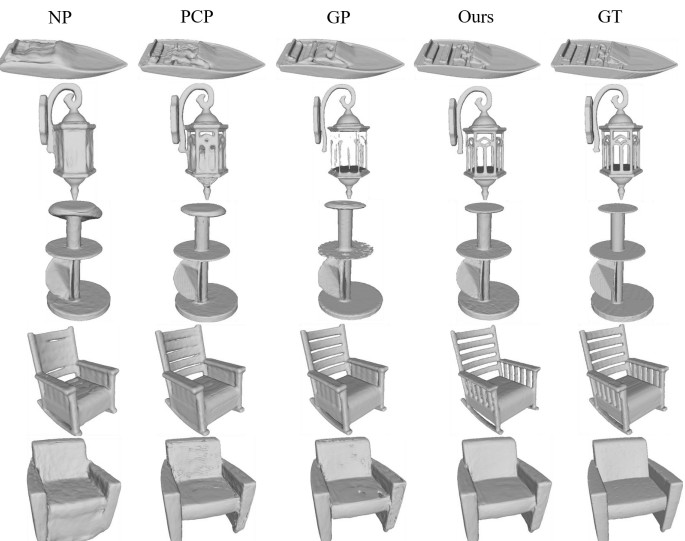

Figure 4: Visual comparison of reconstructions on ShapeNet.

**FAMOUS**. We evaluate the performance of our method on the FAMOUS dataset according to the experimental settings of PCP [64] and NP [21]. Our method demonstrates superiority over recent

approaches, including GP [23], PCP [64], GenSDF [65], FGC [66], NP [21], and IGR [57]. As shown in Tab. 2, we compared the recent methods using $CD_{L2}$ and NC metrics, and our method exhibits outstanding performance. To demonstrate the effectiveness of our method in reconstruction accuracy, we visualize the error-map for comparison in Fig. 5. Compare to the the state-of-art methods, our method has better overall reconstruction accuracy (bluer).

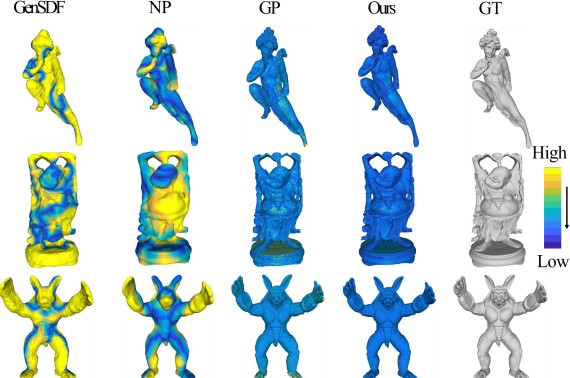

Table 2: Reconstruction accuracy on FA-MOUS in terms of $CD_{L2}$ and NC.

| Methods | $CD_{L2} \times 100$ | NC |
|---------|---------------------|-------|
| IGR | 1.65 | 0.911 |
| GenSDF | 0.668 | 0.909 |
| NP | 0.220 | 0.914 |
| FGC | 0.055 | 0.933 |
| PCP | 0.044 | 0.933 |
| GP | 0.040 | 0.945 |
| Ours | **0.035** | **0.953** |

Figure 5: Visual comparison of error maps on FA-MOUS.

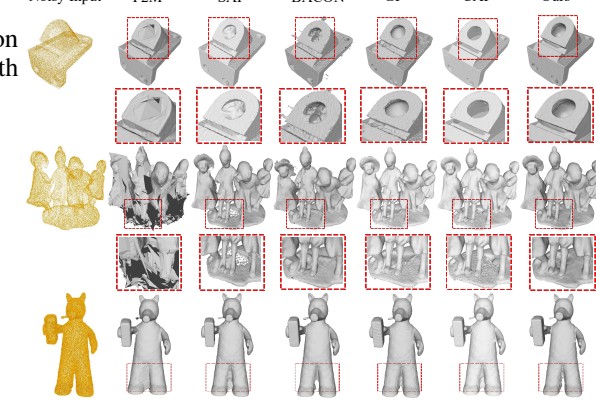

Table 3: Reconstruction accuracy on SRB in terms of $CD_{L1}$ and F-Score with a threshold of 0.01.

| Methods | $CD_{L2} \times 100$ | NC |
|---------|---------------------|-------|
| P2M | 0.116 | 64.8 |
| SAP | 0.076 | 83.0 |
| NP | 0.106 | 79.7 |
| BACON | 0.089 | 82.7 |
| CAP | 0.073 | 84.5 |
| GP | 0.070 | 85.1 |
| Ours | **0.068** | **85.7** |

Figure 6: Visual comparison on SRB.

**SRB**. We validate our method on the real scanned dataset SRB, following the experimental settings of VisCo [67] and GP [23]. In Tab. 3, we compared our approach with recent methods including P2M [68], SAP [56], NP [21], BACON [15], CAP [55], GP [23]. We use $CD_{L1}$ and F-Score to evaluate performance , and we surpass all others in terms of these metrics. As depicted in Fig. 6, our method excels in reconstructing more complete and smoother surfaces.

**D-FAUST**. We evaluate our method on the D-FAUST dataset with SAP [56] settings. As indicated in Tab. 4, we compared our approach with recent methods including IGR [57], SAP [56], GP [23]. Our method excels in $CD_{L1}$, F-Score and NC. As illustrated in Fig. 7, compared to other methods, our approach demonstrates superior accuracy in recovering human body shapes.

**Thingi10K**. We assess the performance of our approach on the Thingi10K dataset, following the experimental setup of SAP [56]. We compared our approach with recent methods including IGR [57], SAP [56], BACON [15], GP [23]. As indicated in Tab. 5, our method surpasses existing methods across in $CD_{L1}$, F-Score and NC metrics. As illustrated in Fig. 8, our method can reconstruct surfaces with more accurate details.

Table 4: Reconstruction accuracy under D-FAUST in terms of $CD_{L1}$ and F-Score with a threshold of 0.01.

| Methods | $CD_{L1}$ | F-Score$^{0.01}$ | NC |
|---|---|---|---|
| IGR | 0.235 | 0.805 | 0.911 |
| SAP | 0.043 | 0.966 | 0.959 |
| GP | 0.015 | 0.975 | 0.978 |
| Ours | **0.009** | **0.986** | **0.988** |

Table 5: Reconstruction accuracy under Thingi 10K in terms of $CD_{L1}$ and F-Score with a threshold of 0.01.

| Methods | $CD_{L1}$ | F-Score$^{0.01}$ | NC |
|---|---|---|---|
| IGR | 0.440 | 0.505 | 0.692 |
| SAP | 0.054 | 0.940 | 0.947 |
| BACON | 0.053 | 0.946 | 0.961 |
| GP | 0.051 | 0.948 | 0.965 |
| Ours | **0.048** | **0.953** | **0.968** |



Figure 7: Visual comparison on D-FAUST.

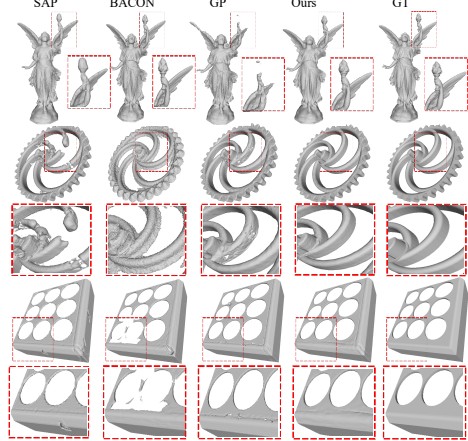

Figure 8: Visual comparison on Thingi10K.

## 4.2 Surface Reconstruction for Real-Scan Scenes

**Datasets and Metrics**. For the scene reconstruction task, we validate our method on the 3DScene [69] and KITTI [70] datasets to assess the performance on large-scale datasets. We keep the same evaluation metrics as those used for shape reconstruction in Sec. 4.1.

**3DScene**. In accordance with the experimental settings of PCP [64], we compared our approach with recent methods including ConvOcc [39], NP [21], PCP [64] and GP [23]. We report the evaluation results of $CD_{L1}$, $CD_{L2}$ and NC, and compared our method with the latest approaches listed in Tab. 6. As illustrated in Fig. 9, our method outperforms prior-based methods and overfitting based methods.

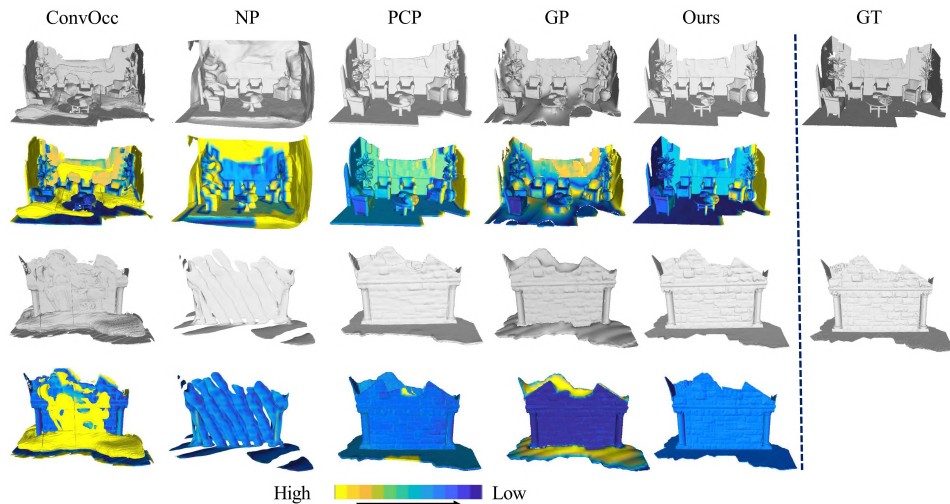

Figure 9: Visual comparison of CD error maps on 3DScene.

Table 6: Reconstruction accuracy on 3DScene in terms of $CD_{L1}$, $CD_{L2}$ and NC.

| Methods | $CD_{L2} \times 100$ | $CD_{L1}$ | NC |
|---------|---------------------|-----------|-----|
| ConvOcc | 13.32 | 0.049 | 0.752 |
| NP | 8.350 | 0.0194 | 0.713 |
| PCP | 0.11 | 0.007 | 0.886 |
| GP | 0.10 | **0.006** | 0.903 |
| Ours | **0.094** | **0.006** | **0.918** |

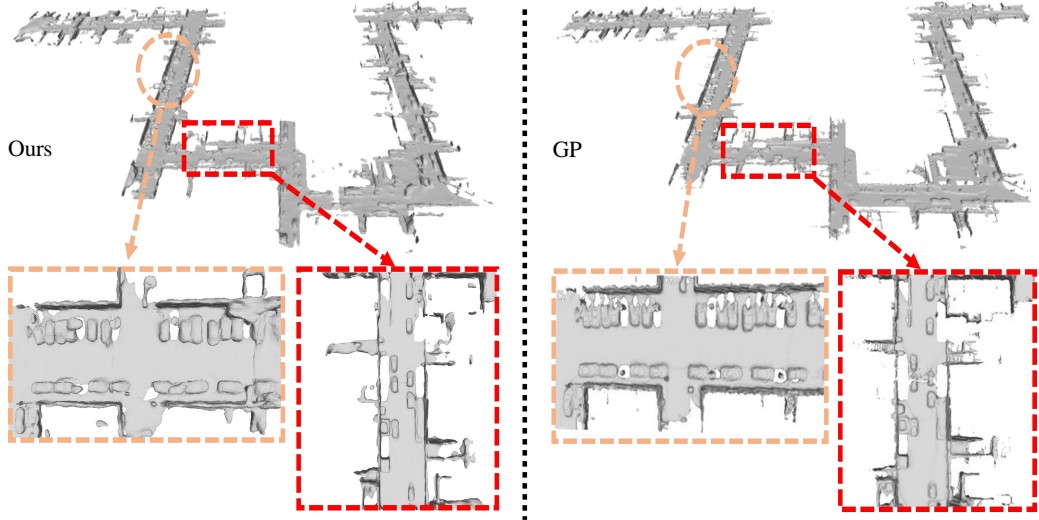

Figure 10: Visual comparison on KITTI.

**KITTI**. We validate our method on the large-scale scanned point cloud dataset KITTI [70], which contains 13.8 million points. As shown in Fig. 10, our approach is capable of reconstructing more complete and accurate surfaces compared to the GP method [23]. GP struggles to reconstruct continuous surfaces such as walls and streets, whereas our method achieves a more detailed reconstruction of objects at various scales in real scanned scenes. It demonstrates that our method is robust when handling point cloud with various scales.

### 4.3 Ablation Experiments

**Effect of Frequency Layers**. We denote the $j$-th layer of the frequency network as $L_j$, a specific combination of frequency feature layers can be formulated as $\{L_i, L_j, L_k\}$, where $\{i, j, k\} \in [1, N_L - 1]$. We evaluate the effectiveness of the frequency transformer layers in Tab. 7 with $CD_{L2}$ and NC, replacing the frequency network with linear layers results in a decrease in the performance of the $CD_{L2}$ and NC metrics. The performance of using only one layer($L_4$) surpasses linear layers. With the increase of the frequency layers, $\{L_4, L_6, L_8\}$ produces best results.

Table 7: Effect of frequency features.

| Layer | $CD_{L2} \times 100$ | NC |
|-------|---------------------|-----|
| Linear | 0.042 | 0.920 |
| $L_4$ | 0.040 | 0.926 |
| $L_4, L_6$ | 0.037 | 0.933 |
| $L_4, L_6, L_8$ | **0.036** | **0.948** |

**Effect of MSP Module**. We report comparisons with different features in Tab. 8. The 'Layer' column denotes the combination of frequency features obtained by the FFT module. For instance, $\{L_4, L_6, L_8\}$ represent the frequency features from the $4^{th}$, $6^{th}$, and $8^{th}$ layers guiding the pulling of the query point in the MSP network, respectively. We find that the accuracy of the network increases with the number of steps. After considering both performance metrics and time efficiency, we have set Step=3 by default.

Table 8: Effect of MSP Module.

| Step | $CD_{L2} \times 100$ | NC |
|------|----------------------|-------|
| 1 | 0.040 | 0.926 |
| 2 | 0.037 | 0.933 |
| 3 | 0.036 | 0.948 |
| 4 | 0.036 | 0.942 |
| 5 | **0.0357** | **0.955** |

**Effect of Loss Functions**. We compared $CD_{L2}$ metric under different loss strategies in Tab. 9. As shown in the table, Weighting query points at different scales effectively enhances reconstruction accuracy and the reconstruction loss allows the network to obtain a complete shape with local details. Furthermore, The gradient loss improves the surface continuity of the object. And the surface supervision loss facilitates the learning of more precise zero-level sets, which also improves the accuracy.

Table 9: Effect of loss functions.

| Loss | $CD_{L2} \times 100$ | NC |
|------|----------------------|-------|
| $\mathcal{L}_{\text{pull}}$ | 0.0443 | 0.937 |
| $\mathcal{L}_{\text{recon}}$ | 0.0383 | 0.946 |
| $\mathcal{L}_{\text{recon}} + \mathcal{L}_{\text{sim}}$ | 0.0367 | 0.948 |
| $\mathcal{L}_{\text{recon}} + \mathcal{L}_{\text{sim}} + \mathcal{L}_{\text{sdf}}$ | **0.0352** | **0.954** |

**Effect of Different Levels of Noise**. We evaluate the reconstruction performance of our method on the Famous dataset under two levels of noise: Mid-Level and Max-Level noise. As shown in Tab. 10, our method outperforms the majority of approaches even in the presence of noisy inputs.

Table 10: Effect of different levels of noise.

| Noise level | NP | PCP | GP | Ours |
|-------------|-------|-------|-------|-----------|
| Mid-Noise | 0.280 | 0.071 | 0.044 | **0.037** |
| Max-Noise | 0.310 | 0.298 | 0.060 | **0.058** |

## 5    Conclusion

We propose a novel method to learn detailed SDFs by pulling queries onto the surface at multi-step. We leverage the multi-level features to predict signed distances, which recovers high frequency details. Through optimization, our method is able to gradually restore the coarse-to-fine structure of reconstructed objects, thereby revealing more geometry details. Visual and numerical comparisons show that our approach demonstrates competitive performance over the state-of-the-art methods.

## 6    Acknowledgment

This work was supported by National Key R&D Program of China (2022YFC3800600), and the National Natural Science Foundation of China (62272263, 62072268), and in part by Tsinghua-Kuaishou Institute of Future Media Data.

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

# A   Implementation Details.

Our network consists of two main parts: frequency feature transformation and multi-step pulling modules in Fig 2 (a) and (b), respectively. For frequency feature transformation, we transform the raw point cloud into frequency features $M$, where $M$ is initialized to 256. Same for the multi-step pulling module, we train a linear sequence neural network (**LSNN**) with shared parameters and we fix intermediate layer output dimension to 512. In the construction of query points, we establish the corresponding pairs between query points and their nearest points on surfaces. Specifically, we follow NP [21] to construct 40 queries for each point of the point cloud, the construction of these query points follows a Gaussian distribution. During the reconstruction process, we use the Marching Cubes algorithm [27] to extract the mesh surface.

During the training process, we do optimization in 40,000 iterations, with an average time of 24 minutes for single-object reconstruction. We utilize a single NVIDIA RTX-3090 GPU for both training and testing.

# B   Additional Experiments

**Effect of Frequency Features.** To further validate the superiority of frequency features, we exclude multi-step pulling, and only use single-layer frequency feature for performance verification against linear layers. We note the frequency feature in the $i$-th layer as $L_i$. We compared the $CD_{L2}$ and NC of specific layers($L_2$, $L_4$, $L_6$, $L_8$) with the linear layers. As shown the Tab. 11 the performance of frequency features at different layers is superior to the linear layers, and with an increase in the number of layers, higher-level frequency conditions enhance the network's performance.

Table 11: Effect of frequency features.

| Layer | $CD_{L2} \times 100$ | NC |
|-------|----------------------|-------|
| Linear | 0.042 | 0.920 |
| $L_4$ | 0.040 | 0.926 |
| $L_6$ | 0.038 | 0.931 |
| $L_8$ | **0.037** | **0.935** |

**Effect of Initialization Strategies**. We compared our initialization strategy with random initialization and MFN-based method (BACON [15]) as example. We compared the metrics of these initialization methods in Tab. 12, which shows that combining random or BACON initialization with our approach does not yield satisfactory results.To further demonstrate the advantages of our initialization method, we visually compared SDF with random initialization and BACON initialization strategies. As shown in the Fig. 11, our method significantly outperforms other initialization methods in terms of convergence speed. In addition, our reconstruction results also indicate that a reasonable initialization method can enable the network to learn more accurate signed distance field. We compared the results with the same iterations and the final results under the default settings for different methods (Final) in Fig. 12, these failed reconstructions based on MFN demonstrate the instability of parameter initialization.

Table 12: Effect of initialization strategies.

| Initialization | $CD_{L2} \times 100$ | NC |
|----------------|----------------------|-------|
| Random | 0.042 | 0.938 |
| BACON | 0.038 | 0.946 |
| Ours | **0.035** | **0.950** |

**Effect of Parameters on Networks.** We compared the parameter quantities of the methods listed in Tab. 13 below. It shows that the parameter number of PCP[64] is the largest among all the three methods, while NP[21] has the least parameters. To further investigate the performance of networks with the similar amount of parameters, we increase the parameters of NP and MultiPull to match PCP. The comparison in the Tab. 14 indicates that both NP and MultiPull show the improved performance. This demonstrates that more parameters are beneficial to improve the performance, but

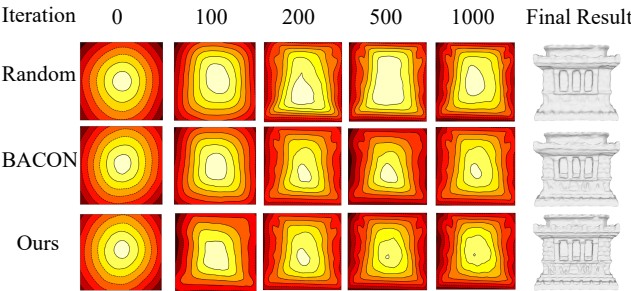

Figure 11: Comparison of signed distances in optimization

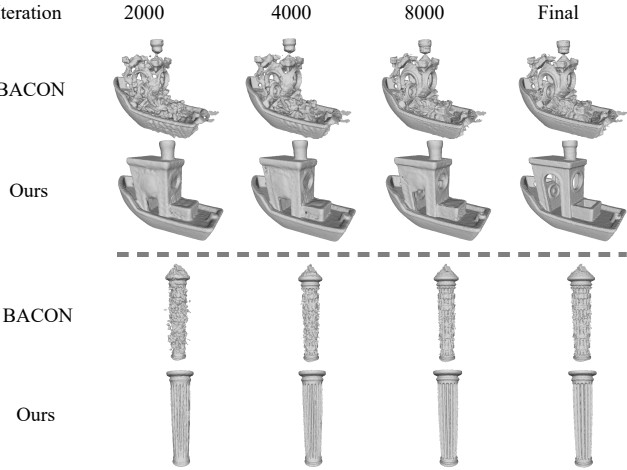

Figure 12: Comparison of the initialization strategies in optimization.

our performance comes from our novel methods rather than more parameters. Meanwhile,we further evaluate inference time with NP and PCP.As shown in Tb. 15 the local based method (PCP) has the longer inference time than the global based method, with NP having the fastest inference time.

Table 13: Comparison at different parameter levels.

| Method | Parameters | $CD_{L2} \times 100$ | NC |
|--------|-----------|--------|-----|
| NP | 1843708 | 0.220 | 0.914 |
| PCP | **7894022** | 0.044 | 0.933 |
| Ours | 2648094 | **0.035** | 0.953 |

Table 14: Comparison at a uniform parameter levels.

| Method | Parameters | $CD_{L2} \times 100$ | NC |
|--------|-----------|--------|-----|
| NP | 7907441 | 0.1966 | 0.917 |
| PCP | **7894022** | 0.044 | 0.933 |
| Ours | 7856620 | **0.0317** | 0.957 |

Table 15: Comparison of Inference Time.

| | PCP | NP | Ours |
|--------|-----|-----|------|
| Inference Time(min) | 0.17 | **0.138** | 0.141 |

**Design of $\mathcal{L}_{\mathbf{grad}}$.** We further discuss the design of $\mathcal{L}_{\mathrm{grad}}$ and effectiveness on performance. We use minimum (min) as the baseline and compare it with using the average (avg). As shown in Tab. 16, using $L_{\mathrm{grad}}(\mathrm{avg})$ to calculate the similarity of query points at different time steps results in a slight increase in CD error. In contrast, $L_{\mathrm{grad}}(\mathrm{min})$ achieves a similar level of similarity but better performance in CD metrics. Therefore, we calculate minimum of the gradient similarities as a constraint to prevent significant deviations in the moving direction during training, making the network more sensitive to changes in gradient direction.

Table 16: Comparison at a uniform parameter levels.

| | w/o $\mathcal{L}_{\mathrm{grad}}$ | $\mathcal{L}_{\mathrm{grad}}(\mathrm{avg})$ | $\mathcal{L}_{\mathrm{grad}}(\mathrm{min})$ |
|---|---|---|---|
| $\mathrm{CD}_{L2} \times 100$ | 0.0388 | 0.0383 | **0.0352** |
| NC | 0.945 | 0.950 | **0.954** |

**Feature Comparison.** We combine MSP with the linear layers and the traditional feature learning encoder PointMLP to explore the effectiveness of MSP. We present the results of combining different feature learning networks with MSP methods in Tab. 17. We denote the single moving operation in MSP as Pull and use multiple feature learning networks as baselines. Due to the lack of an FFT module, the same features are used for multiple offsets in linear+MSP and PointMLP + MSP. As shown in Tab. 17, the combination of different feature extraction networks and MSP achieved better results in terms of both CD and NC metrics. We further demonstrate the effectiveness of MSP in Fig. 13 and Fig. 14 in the PDF. PointMLP / linear+MSP can generate finer local details compared to PointMLP / linear + Pull.

Table 17: Comparison of Reconstruction Accuracy in $\mathrm{CD}_{L2} \times 100$

| Iteration | 10K | 20K | 40K |
|---|---|---|---|
| Linear+Pull | 0.078 | 0.061 | **0.055** |
| PointMLP+Pull | 0.064 | 0.045 | **0.037** |
| Linear+MSP | 0.073 | 0.058 | **0.041** |
| PointMLP+MSP | **0.053** | **0.042** | **0.037** |

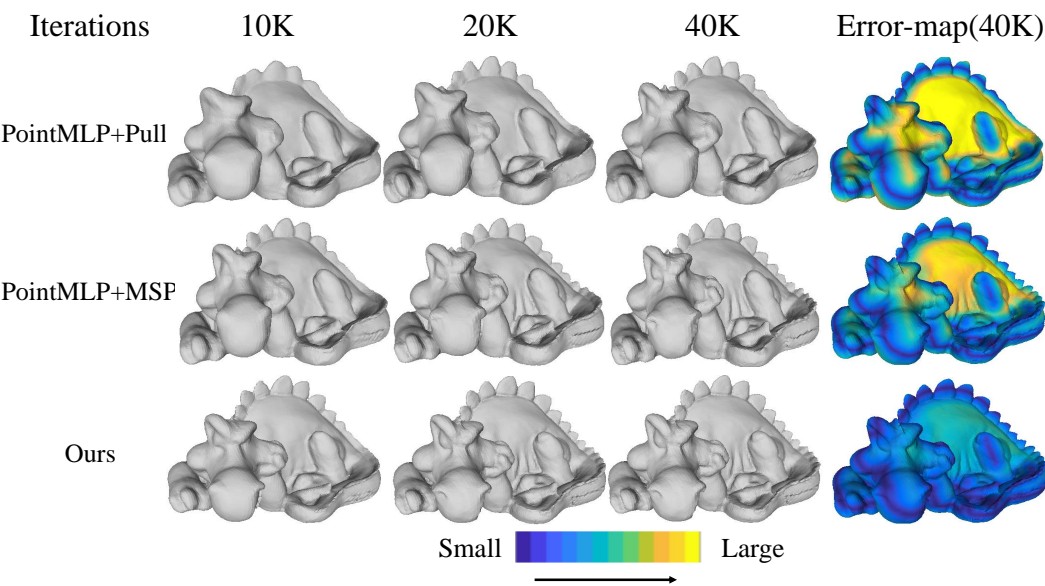

Figure 13: Comparison of different feature encoders and MSP module on FAMOUS dataset.

**Default setting of Step.** We set Step to 3 by default for two reasons: (1) MSP can bring partial accuracy improvement when learning more steps, but more network parameters are also needed. We

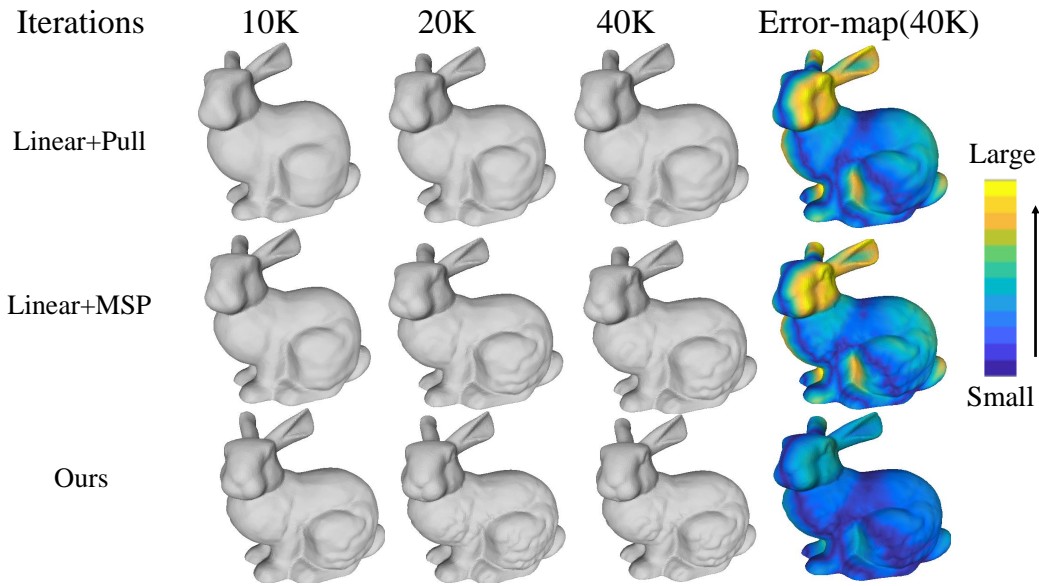

Figure 14: Comparison of different feature encoders and MSP module on FAMOUS dataset.

present comparisons under different steps in Tab. 8. (2) Although deeper frequency features can represent more comprehensive information, combining frequency features from different layers can achieve better results. We denote the combination of the frequency. We conduct the experiment according to the default settings, and the results are shown in Tab. 18.

Table 18: Comparison of $CD_{L2} \times 100$, NC under different feature combination.

|  | $L_4, L_4, L_4$ | $L_6, L_6, L_6$ | $L_8, L_8, L_8$ | $L_4, L_6, L_8$ |
|---|---|---|---|---|
| $CD_{L2} \times 100$ | 0.0417 | 0.0397 | 0.0360 | **0.0343** |
| NC | 0.922 | 0.927 | 0.938 | **0.948** |

## C Additional Results

**Comparison Details for ShapeNet**. The complete comparison under all the five scenes of the ShapeNet dataset. The results are shown in Tab. 19 to Tab .21. We use Chamfer Distance ($CD_{L1}, CD_{L2}$) and NC as evaluation metrics.

Table 19: Reconstruction accuracy on ShapeNet in terms of $CD_{L2}$

| Class | ATLAS | DSDF | NP | PCP | GP | Ours |
|---|---|---|---|---|---|---|
| Display | 1.094 | 0.317 | 0.039 | 0.0087 | 0.0082 | **0.0074** |
| Lamp | 1.988 | 0.955 | 0.080 | 0.0380 | 0.0347 | **0.0301** |
| Airplane | 1.011 | 1.043 | 0.008 | 0.0065 | 0.0007 | **0.0006** |
| Cabinet | 1.611 | 0.921 | 0.026 | 0.0153 | 0.0112 | **0.0105** |
| Vessel | 0.997 | 1.254 | 0.022 | 0.0079 | 0.0033 | **0.0028** |
| Table | 1.311 | 0.660 | 0.060 | 0.0131 | 0.0052 | **0.0047** |
| Chair | 1.575 | 0.483 | 0.054 | 0.0110 | 0.0043 | **0.0036** |
| Sofa | 1.307 | 0.496 | 0.012 | 0.0086 | 0.0015 | **0.0009** |
| Mean | 1.368 | 0.766 | 0.038 | 0.0136 | 0.0086 | **0.0075** |

Table 20: Reconstruction accuracy on ShapeNet in terms of NC.

| Class | ATLAS | DSDF | NP | PCP | GP | Ours |
|---|---|---|---|---|---|---|
| Display | 0.828 | 0.932 | 0.964 | 0.9775 | 0.9847 | **0.9855** |
| Lamp | 0.593 | 0.864 | 0.930 | 0.9450 | 0.9693 | **0.9710** |
| Airplane | 0.737 | 0.872 | 0.947 | 0.9490 | 0.9614 | **0.9633** |
| Cabinet | 0.682 | 0.872 | 0.930 | 0.9600 | 0.9689 | **0.9693** |
| Vessel | 0.671 | 0.841 | 0.941 | 0.9546 | 0.9667 | **0.9671** |
| Table | 0.783 | 0.901 | 0.908 | 0.9595 | 0.9755 | **0.9758** |
| Chair | 0.638 | 0.886 | 0.937 | 0.9580 | 0.9733 | **0.9757** |
| Sofa | 0.633 | 0.906 | 0.951 | 0.9680 | 0.9792 | **0.9822** |
| Mean | 0.695 | 0.884 | 0.939 | 0.9590 | 0.9723 | **0.9737** |

Table 21: Reconstruction accuracy on ShapeNet in terms of F- Score with thresholds of 0.002 and 0.004.

| | F-Score$^{0.002}$ | | | | | | F-Score$^{0.004}$ | | | | | |
|---|---|---|---|---|---|---|---|---|---|---|---|---|
| | ATLAS | DSDF | NP | PCP | GP | Ours | ATLAS | DSDF | NP | PCP | GP | Ours |
| Display | 0.071 | 0.632 | 0.989 | 0.9939 | 0.9963 | **0.9966** | 0.179 | 0.787 | 0.991 | 0.9958 | 0.9963 | **0.9980** |
| Lamp | 0.029 | 0.268 | 0.891 | 0.9382 | 0.9455 | **0.9473** | 0.077 | 0.478 | 0.924 | 0.9402 | 0.9538 | **0.9561** |
| Airplane | 0.070 | 0.350 | 0.996 | 0.9942 | 0.9976 | **0.9989** | 0.179 | 0.566 | 0.997 | 0.9972 | 0.9989 | **0.9991** |
| Cabinet | 0.077 | 0.573 | 0.980 | 0.9888 | 0.9901 | **0.9913** | 0.195 | 0.694 | 0.989 | 0.9939 | 0.9946 | **0.9969** |
| Vessel | 0.058 | 0.323 | 0.985 | 0.9935 | 0.9956 | **0.9962** | 0.153 | 0.509 | 0.990 | 0.9958 | 0.9972 | **0.9979** |
| Table | 0.080 | 0.577 | 0.922 | 0.9969 | 0.9977 | **0.9982** | 0.195 | 0.743 | 0.973 | 0.9958 | **0.9990** | 0.9988 |
| Chair | 0.050 | 0.447 | 0.954 | 0.9970 | 0.9979 | **0.9980** | 0.134 | 0.665 | 0.969 | 0.9991 | 0.9990 | **0.9994** |
| Sofa | 0.058 | 0.577 | 0.968 | 0.9943 | 0.9974 | **0.9981** | 0.153 | 0.734 | 0.974 | 0.9987 | **0.9992** | **0.9992** |
| Mean | 0.062 | 0.212 | 0.961 | 0.9871 | 0.9896 | **0.9906** | 0.158 | 0.717 | 0.976 | 0.9899 | 0.9923 | **0.9932** |

**Comparison Details for 3DScene**. We also provide detailed metrics for single scenes in 3DScene dataset. We evaluate it by $CD_{L1}$, $CD_{L2}$ and NC. As shown in the Tab. 22, our approach achieves the best performance across all scenes.

Table 22: $CD_{L1}$, $CD_{L2} \times 100$ and NC comparison under 3DScene.

| | | ConvOcc | NP | PCP | GP | Ours |
|---|---|---|---|---|---|---|
| Burghers | $CD_{L2} \times 100$ | 26.69 | 1.76 | 0.267 | 0.246 | **0.228** |
| | $CD_{L1}$ | 0.077 | 0.010 | **0.008** | **0.008** | **0.008** |
| | NC | 0.865 | 0.883 | 0.914 | 0.926 | **0.934** |
| Lounge | $CD_{L2} \times 100$ | 8.68 | 39.71 | 0.061 | 0.055 | **0.048** |
| | $CD_{L1}$ | 0.042 | 0.059 | 0.006 | **0.005** | **0.005** |
| | NC | 0.857 | 0.857 | 0.928 | 0.922 | **0.949** |
| Copyroom | $CD_{L2} \times 100$ | 10.99 | 0.051 | 0.076 | 0.069 | **0.064** |
| | $CD_{L1}$ | 0.045 | 0.011 | 0.007 | **0.006** | **0.006** |
| | NC | 0.848 | 0.884 | 0.918 | 0.929 | **0.923** |
| Stonewall | $CD_{L2} \times 100$ | 19.12 | 0.063 | 0.061 | 0.058 | **0.043** |
| | $CD_{L1}$ | 0.066 | 0.007 | 0.0065 | 0.006 | **0.005** |
| | NC | 0.866 | 0.868 | 0.888 | 0.893 | **0.936** |
| Totepole | $CD_{L2} \times 100$ | 1.16 | 0.19 | 0.10 | 0.093 | **0.087** |
| | $CD_{L1}$ | 0.016 | 0.010 | 0.008 | **0.007** | **0.007** |
| | NC | 0.325 | 0.765 | 0.784 | 0.847 | **0.851** |

**Computational Complexity**. We report our computational complexity in Tab. 23, we present numerical comparisons with the latest overfitting-based methods, including NP and PCP, using different point counts, such as 20K and 40K. The benchmark rounds for both NP and PCP are set at 40K. NP does not require learning priors, resulting in the highest operational efficiency. PCP needs to learn priors, which requires additional time. To achieve more refined results, we dedicate extra time to learning the frequency features of point clouds and computing the sampling point strides. Consequently, our speed is slower compared to NP. However, it is noteworthy that our method outperforms PCP and operates faster even without using local priors.

Table 23: Comparison of computational complexity.

| Time/GPU Memory | 20K | 40K |
|---|---|---|
| NP | **12min/1.5G** | **15min/2.3G** |
| PCP | 14min/1.9G | 18min/2.7G |
| Ours | 13min/1.8G | 16min/2.5G |

## D   Limitation

We propose a method that approximates the accurate signed distance field through multi-step optimization, achieving more precise results. However, there is still room for further optimization in terms of time and computational efficiency as shown in Tab. 8 and Tab. 14. In future work, we will continue to explore how to integrate multi-resolution (such as NGLOD [16] and Instant-NGP [71]) features effectively to balance computational efficiency and accuracy.

