# OpenReview forum: "MultiPull: Detailing Signed Distance Functions by Pulling Multi-Level Queries at Multi-Step"
_NeurIPS.cc/2024/Conference — NeurIPS 2024 poster_

### Official Review · Reviewer_VUUa · 2024-07-10

**Soundness:** 2
**Presentation:** 2
**Contribution:** 3
**Rating:** 5
**Confidence:** 4

**Summary:**

The authors propose a method to reconstruct neural multi-scale SDFs from point clouds using an optimization-based approach. The multi-scale SDF is represented with a network architecture based on previous work [11] and optimized using a new iterative pulling approach, where the SDF for each scale is optimized to pull a query point to the 0-level-set (using the gradient and value of the SDF at the query), starting with a relatively distant query point for the coarsest SDF, and using the updated query position from the result of the previous scale in the next-finer SDF. Losses based on GridPull [20], but adapted to the multi-scale setting are used as objective for the optimization. The authors show that this approach performs better than several exisiting surface reconstruction methods on several synthetic and real-world datasets.

**Strengths:**

- The iterative multi-scale pulling approach seems novel as far as I can tell, and does make sense to me (apart from some details). It effectively optimizes coarser SDFs to describe regions more distant from the surface but to miss details close to the surface, and more detailed SDFs to describe regions close to the surface with more details.
- The evaluation is quite comprehensive, including several datasets and a good ablation study (although some relevant recent methods are not included), and shows a good improvement over existing work.

**Weaknesses:**

- The evaluation is missing some relevant recent methods for optimization-based surface reconstruction. (See details below)
- Some relevant related work is not cited. (See details below)
- The exposition is missing motivation for some design choices and is hard to understand in some parts. (See details below)

Overall, even though the authors missed some related work, results look quite promising and I think the idea is novel and interesting enough to make me lean towards acceptance. The exposition can probably be improved somewhat in the final version.

Details:

- The field of surface reconstruction from point clouds is quite vast, so the authors missed several works, for example (see also related work within these papers to get more missing work):
  -Training-free methods:
    - Neural-Singular-Hessian: Implicit Neural Representation of Unoriented Point Clouds by Enforcing Singular Hessian, Wang et al., TOG 2023
    - Iterative Poisson Surface Reconstruction (iPSR) for Unoriented Points, Hou et al., TOG 2022
  - Learning-based methods:
    - 3DShape2Vecset: A 3D Shape Representation for Neural Fields and Generative Diffusion Models, Zhang et al., TOG 2023
    - Geoudf: Surface reconstruction from 3d point clouds via geometry-guided distance representation, Ren et al., TOG 2023
  - Given good normals, surface reconstruction becomes much easier - the normal computation could be followed by Poisson reconstruction based on the normals for example. Therefore papers to compute oriented normals are relevant, such as:
    - Orienting Point Clouds with Dipole Propagation, Metzer et al., TOG 2021
    - Globally Consistent Normal Orientation for Point Clouds by Regularizing the Winding-Number Field, Xu et al., TOG 2023
    - SHS-Net: Learning Signed Hyper Surfaces for Oriented Normal Estimation of Point Clouds, Li et al., CVPR 2023

- Several design choices are not motivated well enough:
  - It is unclear to me why the SDF evaluated at Q_1 uses a feature y_1 as input that was computed from a query position Q_0, rather than a feature computed at query position Q_1. This is an interesting design choice that could use more discussion. An ablation would also be interesting: how would the method perform when re-evaluating h_0, ...,  h_i at Q_i instead of Q_0 when computing y_i for f(Q_i, y_i) (in that case inputting Q_i a second time to the LSNN might also not be necessary)?
  - In Eq. 6, the design choice of only scaling D_1 and D_2 with the error-based factor, but not D_3 needs to be motivated. Also, as Section 3.2 was written for a general number of layers N_L, it would be good to continue this generality in Section 3.3, if possible. How would Eq. 6 look for a general N_L., for example?
  - In Eq. 7, the design choice for using the minimum over the three levels.

- The exposition is hard to understand in some parts and could be improved:
  - Several passages are currently hard to understand and could use clarification. For example: "The is because Q_i  locate at different scales in space,, making it difficult to simultaneously account for the optimization step sizes at different scales with the inconsistent gradient directions caused by continuous movement." Its possible to get a good guess at what the authors mean here, but it requires re-reading a few times and is not 100% clear even then. Partially this seems to be due to bad grammar, partially due to a somewhat convoluted formulation. Generally the paper text could use more passes to avoid these issues.
  - In Section 3.1, it should be mentioned early on that the FFT module is based on (or at least very similar to) MFN [11]. Also, the exact differences between the proposed approach in Section 3.1 and the approach of MFN [11] should be discussed and clarified.
  - In Section 3.3, it would be good to mention that the loss functions are based on GridPull [20]. Also, it would be good to discuss the exact differences to GridPull (I assume the adpation to the multi-scale setting is the main difference).
  - The architecture of the Linear Sequence Neural Network (LSNN) is never described. If it is taken from existing work, this work should be cited, otherwise it needs to be described, at least in Appendix A.
  - In the paragraph starting at line 112 and Eq. 4, it is unclear which weights are initialized according to this strategy (the omega_i in Eq. 1? The W_i in Eq. 3? The offset theta_i and b_i?), and which parameters are plotted in Figure 3 (the L_i are not defined clearly).
  - In Eq. 4, \eta is not defined.
  - Near line 138, the query point update should probably use Q_{i-1} everywhere on the right-hand-side (instead of Q_i, which is not known yet). Or alternatively, Q_{i+1} on the left-hand side.
  - In Eq. 1, I don't follow why the predicted signed distance f(Q_1, y_1) is in the loss. Why should that be optimized to be 0? This should be clarified.
  - in Eq. 1, ", i \in [1,I]" can be removed, since it is redundant with the subscript in the sum
  - In the section starting at line 160, how is the target surface point q determined? This is described in the appendix, but should be shortly mentioned in the main paper as well.
  - In Eq. 7, argmin should probably be min instead. Also, the "1 - ..." in both lines seems odd, should it only be in one of the lines (otherwise the loss would be *higher* the more similar the gradient directions are, instead of lower).
  - Near Eq. 8, the authors mention that L_surf encourages the SDF at the *final* query point to be close to 0, but Eq. 8 does not show an index for Q. Is L_surf applied to Q_{N_L} only or to all Q_i? This should be clarified.
  - Near line 202: Fig. 4.1 should probably be  Fig. 5
  - In Table 7, Step 5 should probably have CD_L2 of 0.0357 (not 0.357)
  - SAL [15] is in the bibliography, but not referenced from Section 2.
  - Citations [45] and [59] are duplicates.

- An inference time comparison between methods should be included.

**Questions:**

A clarification of the design choices could help, as well as a discussion of the differences and advantages/disadvantages compared to the missing related work.

**Limitations:**

Limitations in the long computation time are discussed. I don't foresee any particular negative societal impacts.

---

> ### Author Rebuttal · Authors · 2024-08-06
>
> Thank you for your acknowledgements in our method and results. In the following, we respond to the comments with respect to the weaknesses and questions in turn. All experiments presented in the rebuttal are based on the FAMOUS dataset.
>
> ##### **(1) Further comparison with related works.**
>
> > The field of surface reconstruction from point clouds is quite vast, so the authors missed several works
>
> **RESPONSE**: We follow the reviewer's suggestion to make a further comparison of the related works. Due to the time limit, we just select overfitting based method Neural-Singular-Hessian [1] for visual comparison, and we will add additional comparison results of other work in our version. As shown in Fig. C in the PDF, our method still has competitive performance compared to Neural-Singular, and we perform better in generating local details.
>
> ##### **(2) Concern about frequency features.**
>
> > It is unclear to me why the SDF evaluated at $Q_1$ uses a feature $y_1$ as input that was computed from a query position $Q_0$, rather than a feature computed at query position $Q_1$. An ablation would also be interesting: how would the method perform when re-evaluating $h_0$, ..., $h_i$ at $Q_i$ instead of $Q_0$ when computing $y_i$ for f($Q_i$, $y_i$)
>
> **RESPONSE**: All frequency features y in the FFT are obtained from the query point $Q_0$. we transform the query point $Q_0$ into frequency features as conditions to guide the movement of $Q_0$, $Q_1$, and $Q_2$. The deeper the features in the FFT, the more comprehensive the information contained. We hope to use this movement strategy to enable MSP to learn the global information of the input from low-frequency features, and then optimize local details through high-frequency features. In summary, the frequency features of FFT come from the same set of query points. If we re-evaluate the $Q_i$ of each stage into FFT, it will disrupt the consistency of the features. Therefore, FFT cannot simultaneously accept multiple point clouds for frequency feature encoding.
>
> ##### **(3) Details of $\mathcal{L}_{recon}$.**
>
> >  In Eq. 6, the design choice of only scaling $D_1$ and $D_2$ with the error-based factor, but not $D_3$ needs to be motivated. How would Eq. 6 look for a general $N_L$., for example?
>
> ##### **(4) Design choice for $\mathcal{L}_{grad}$.**
>
> > In Eq. 7, the design choice for using the minimum over the three levels.
>
> **RESPONSE**: In order to maintain consistency in the moving direction of query points at different steps during continuous offset, we naturally choose gradient similarity $L_{grad}$ as a constraint term. Furthermore, we force the network to focus on the step where the gradient direction changes the most. We conduct experiments according to reviewer uTM5 to demonstrate the validity in Tab. A.
>
> **RESPONSE**:  We force network to focus more on outliers and points close to the surface to accelerate network convergence in $L_{recon}$. Therefore, we make the error-based factors of $D_3$ higher than  and $D_1$ and $D_2$. We define a set $D$={$D_{1}, D_{2}, D_{3}$} and its corresponding weights $w$={$\alpha, (1- \alpha )^{\gamma}, 1$}, then $L_{recon}$ can be expressed as: $L_{recon}$ = $\sum^{3}_{i=1}w_i D_i$.
>
> ##### **(5) Inference time comparison.**
>
> > An inference time comparison between methods should be included.
>
> **RESPONSE**: In Tab. G, we further evaluate inference time with NP and PCP. The results show that the local based method (PCP) has the longer inference time than the global based method (NP, Multiplaull), with NP having the fastest inference time.
>
> ##### **(6) Explaination about details.**
>
> > The exposition is hard to understand in some parts and could be improved.
>
> **RESPONSE**: Thank you for the reviewer's suggestions for writing revisions and reminders for adding details. Here we will present the issues that require further discussion, and we will refine their writing suggestions and revision suggestions in the next version.
>
> ##### **Differences of GridPull.**
>
> Our method uses pulling based loss like GP [2]. However, our optimization strategy is different from GP. GP uses grid points to perform trilinear interpolation on point inputs to predict the SDFs, which does not require any neural network for optimization. Instead, We need multi-scale frequency features as conditions to move the query points toward the underlying surface of the point cloud in multiple steps. This process uses LSNN network to learn SDFs and gradients at different steps. As shown in the visual comparison in the main text, optimization based methods can learn smoother surfaces.
>
> ##### **Architecture of LSNN.**
>
> LSNN consists of 8 linear layers, with seven layers having a dimension of 512, and the last layer having a dimension of 1. We use LSNN to predict the signed distance value $f (Q_i, y_i)$ of the query point $Q_i$ under the frequency feature $y_i$ after the $i-th$ step of movement. We will add relevant content in the subsequent appendix. We apologize for any inconvenience caused to your reading.
>
> ##### **Construction of query points.**
>
> We construct query points based on the sampling method of GridPull [2]. Due to space limitations, reviewer can refer to reviewer NaoM's question (6) for a detailed introduction.
>
> ##### **Scope of use of $\mathcal{L}_{surf}$.**
>
> We only use $\mathcal{L}_{surf}$ at $f(Q_3)$ to make the network more confident in predicting the zero-level-set on the point cloud surface.
>
> [1] Wang Z, Zhang Y, Xu R, et al. Neural-Singular-Hessian: Implicit Neural
> Representation of Unoriented Point Clouds by Enforcing Singular
> Hessian[J]. ACM Transactions on Graphics (TOG), 2023, 42(6): 1-14.
>
> [2] Chen C, Liu Y S, Han Z. Gridpull: Towards scalability in learning
> implicit representations from 3d point clouds[C]//Proceedings of the
> ieee/cvf international conference on computer vision. 2023: 18322-18334.

---

> > ### Comment · Reviewer_VUUa · 2024-08-12
> >
> > Thanks for the detailed replies and the new set of evaluations!
> >
> > The added comparison to Neural-Singular-Hessian looks promising. This addresses my concern to some extent (although a more complete comparison would obviously be better).
> >
> > About the frequency features, I understand the author's point of view that choosing the same query point for all features ensures their consistency, but on the other hand, it also requires these features to make predictions about distant more positions, i.e. about query positions Q1, Q2, ..., as well as needing to take the trajectory caused by previous levels into account (i.e. where did Q1 end up given the prediction of the previous level?), which seems like a more difficult non-local task to me. A more in-depth discussion/analysis would be interesting here, but I would consider this to be optional.
> >
> > The author's response addresses most of my concerns.

---

> > > ### Author Response · Authors · 2024-08-12
> > >
> > > Thanks for the comments.
> > >
> > > Actually, we were using multi-step pulling procedure, that is the end point of previous pulling will be the start point of the next pulling. This is like recursively calling the SDF iteration by iteration, which does not bring too much burden. Moreover, only the first pulling is observed the large movement, while all the following pulling is just like little adjustment. This is because the first pulling will pull most of queries to somewhere by the zero-level set, then the following pulling will not have too much room to move the queries, or the queries will get pulled far away from the zero-level set in another direction. Thus, pulling in the last several steps is still happening in a pretty local area.
> > >
> > > Moreover, it did slow down the feature query process if we use pulled queries like Q1 or Q2 to query features. Currently, using Q0 can help us obtain all features needed in the following pretty fast.
> > >
> > > We will make sure to include these as a discussion in our revision.
> > >
> > > Best,
> > >
> > > The authors

---

### Official Review · Reviewer_ZVf1 · 2024-07-11

**Soundness:** 3
**Presentation:** 2
**Contribution:** 2
**Rating:** 5
**Confidence:** 2

**Summary:**

This paper presents a model for reconstructing SDF from point clouds. The proposed approach introduces components, including Frequency Feature Transformation and Multi-Step Pulling, to iteratively refine the reconstructed SDF. The experiments demonstrate that this method performs well across multiple 3D object and scene datasets.

**Strengths:**

The use of multiple levels of features and query is a valid and well-founded idea. The experiments are through and the visualization of the results looks good.

**Weaknesses:**

1. Some intuitions behind the methodology are not clearly explained. Specifically, the rationale for using frequency features to represent query points needs further clarification. While the use of multiple levels of query points is understandable, the choice of LSNN at each level and how query points are used in this model require more detailed explanation.
2. Some of the notations are inconsistent. S is used for both raw point cloud in L91 and SDF surface on L99. D is used for both feature dimensions and distance.
3. Minor: The number reported on the last row of Table 7 appears to be wrong.

**Questions:**

1. What’s the intuition for using frequency features and why using LSNN in MSP?
2. From the visualization, the proposed method seems to produce results with fewer holes and smoother surfaces. Is this due to the model learning better features, or is there a specific loss design different from prior work?
3. Could you compare the MSP using the same features from prior work to demonstrate its performance even without the proposed feature? This would help to convincingly show its effectiveness.

**Limitations:**

No negative social impacts have been identified.
See weakness.

---

> ### Author Rebuttal · Authors · 2024-08-06
>
> Thank you for your acknowledgements in our method and results. In the following, we respond to the comments with respect to the weaknesses and questions in turn. All experiments presented in the rebuttal are based on the FAMOUS dataset.
>
> ##### **(1) Motivation of FFT.**
>
> > The rationale for using frequency features to represent query points needs further clarification.
>
> **RESPONSE**: Previous works fail to accurately represent the local geometry of objects by learning global features. While adopting local methods can alleviate this issue, they typically suffer from high parameter counts and extended training times. Inspired by LOD models, we transform complex features into a more intuitive and interpretable frequency domain representation for neural networks. By utilizing simple frequency domain feature transformations and linear layers (Equation (2) and (3) in the main text ), we can effectively represent features at different scales, it also aids in predicting implicit fields from coarse to fine, where high frequency features help produce more sensitive adjust in signed distance field, leading to high frequency geometry details.
> ##### **(2) Details about LSNN.**
>
> > While the use of multiple levels of query points is understandable, the choice of LSNN at each level and how query points are used in this model require more detailed explanation.
>
> **RESPONSE**: LSNN is a neural network consisting of 8 fully connected layers with ReLU activation functions. The first 7 layers have a dimension of 512, while the last layer has a dimension of 1. We use it to predict the SDFs and gradient of query points at different steps. The FFT module encodes the query point $Q_0$ into frequency features $y$ at different scales. We select three of these frequency features as conditions for the multi-step offset of the MSP module's query point to predict the implicit field $f$ after the offset. We denote the $i$-th step query points and frequency feature as $Q_i$ and $y_i$, respectively. We concatenate $Q_i$ and $y_i$ and use the LSNN network with shared parameters to predict the signed distance $f(Q_i, y_i)$ and the gradient $\nabla f(Q_i, y_i)$. In this way, we finally obtain the different levels of query points $Q_1$, $Q_2$, and $Q_3$.
>
> ##### **(3) Analysis of Results.**
>
> > From the visualization, the proposed method seems to produce results with fewer holes and smoother surfaces. Is this due to the model learning better features, or is there a specific loss design different from prior work?
>
> **RESPONSE**: We analyze the effectiveness of different modules from two aspects: (1) We conduct ablation experiments on modules other than MSP in Tab. E. We first remove the FFT module and use linear layers to learn query point features for reconstruction, resulting in a significant increase in CD error, which proves the effectiveness of the FFT module. Next, we gradually remove different $L_{sim}$and $\mathcal{L}_ {surf} $to explore the importance of these constraint terms. After that, we obtain slightly worse results after removing two losses. (2) We further explore the impact of MSP on accuracy. As shown in Tab. B, the CD error increases significantly when the steps are reduced, and we get the worst result when $Step=1 $. Overall, the offset steps and FFT module have the greatest impact for our method.
>
> ##### **(4) Feature Comparison.**
>
> > Could you compare the MSP using the same features from prior work to demonstrate its performance even without the proposed feature? This would help to convincingly show its effectiveness.
>
> **RESPONSE**: We followed the reviewer's suggestion to combine MSP with the linear layers and the traditional feature learning encoder PointMLP [1] to explore the effectiveness of MSP. We present the results of combining different feature learning networks with MSP methods in Tab. F. We denote the single moving operation in MSP as Pull and use multiple feature learning networks as baselines. Due to the lack of an FFT module, the same features are used for multiple offsets in linear+MSP and PointMLP + MSP. As shown in Tab. F, the combination of different feature extraction networks and MSP achieved better results in terms of both CD and NC metrics. We further demonstrate the effectiveness of MSP in Fig. A and Fig. B in the PDF. PointMLP / linear+MSP can generate finer local details compared to PointMLP / linear + Pull.
>
> ##### **(5) Typo correction.**
>
> > S is used for both raw point cloud in L91 and SDF surface on L99. D is used for both feature dimensions and distance. Minor: The number reported on the last row of Table 7 appears to be wrong
>
> **RESPONSE**: Thank you for the typo errors raised by the reviewer. We will correct these errors in future versions. We apologize for any inconvenience caused during your reading.
>
> [1] Ma X, Qin C, You H, et al. Rethinking network design and local geometry
> in point cloud: A simple residual MLP framework[J]. arXiv preprint
> arXiv:2202.07123, 2022.

---

> > ### Comment · Reviewer_ZVf1 · 2024-08-12
> >
> > Thank you for your reply, I have no further questions at this time.

---

### Official Review · Reviewer_NaoM · 2024-07-12

**Soundness:** 3
**Presentation:** 3
**Contribution:** 3
**Rating:** 7
**Confidence:** 4

**Summary:**

This paper proposes MultiPull, a method for reconstructing a surface model by SDF from a 3D point cloud containing only the coordinates of each point. The paper proposes a method for estimating the SDF using the Fourier transform of the surface model predicted from the point cloud only at multi-scale. It also introduces gradient consistency and distance awareness for multi-scale consistency. The key idea is to use multi-scale frequency features, for which we introduce an FFT module corresponding to an encoder and an MSP module corresponding to an implicit decoder. The proposed method shows high reconstruction performance compared to existing methods and is effective on different datasets.

**Strengths:**

- The problem of reconstructing a surface from only 3D point clouds, accessible in a real environment by a 3D sensor, is a common but important problem and an important technology for real-world applications of 3D data. However, the reconstruction performance of existing methods is limited, and a better method is needed. The proposed method aims to fill this gap, and the significance of the research is highly appreciated.
- The paper is well written, especially in its straightforward and accurate description of the difficulties in setting up the problem. The detailed description of the proposed method also makes the paper easy to understand.
- The insight into the initialization of MFN is an interesting point, which I consider to be one of the contributions of the paper.
- Experimental results on various datasets are presented, showing the effectiveness of the proposed method.

**Weaknesses:**

- l.138: "\cdot" is better than "\times" to avoid confusion with the cross product.
- My understanding is that there seems to be no guarantee that the scale of the displacement of the query point at each frequency varies with the frequency of interest. Is this understanding correct? Is there some possible constraint on the magnitude of the slope that would be consistent with the frequency of interest? (Or is it that this concern does not exist, since the experimental separation by frequency has been verified?)
- l.166: For the purpose of consistency, would it not be appropriate to use the average or correlation rather than argmin to select only one of Q1, Q2, or Q3? If you have any additional explanation, I would appreciate it.
- In the introduction, it would be desirable to specify whether the proposed method is a test-only optimization (i.e., does not require prior training, such as NeuralPull or SALD) or a method that requires prior training (e.g., DeepSDF). Figure 3 in [*1] clearly shows the position of the proposed method in relation to related research. My understanding is that the paper plots to the same point as NGLOD, and it is preferable to explicitly state that only test time is considered. (If my understanding is incorrect, please correct me.)
- l.243: After considering both performance metrics and time efficiency, we have set Step=3 by default.", can the authors discuss this trade-off? Given the nature of the LOD and the trade-off with the 3D geometry compression ratio, there may be future applications related to mesh compression.

[*1] Francis Williams, Zan Gojcic, Sameh Khamis, Denis Zorin, Joan Bruna, Sanja Fidler, Or Litany. Neural Fields As Learnable Kernels for 3D Reconstruction. CVPR2022.

**Questions:**

- l.77: "However, inferring implicit functions without 3D supervision requires a very long convergence process, which limits the performance of unsupervised methods in large-scale point cloud data." I agree with this point from my experiments, but can the author provide a specific reference in the paper?
- l.99: "To this end, we constrain query points to be as close to their nearest neighbor on S." I understood sampling from the nearest neighbor, but did not understand how Q is constructed. Are non-nearest points just ignored?

**Limitations:**

- I agree with the limitations described in the paper. This is a promising direction for future work.
- If there are areas of future work that you would like to see described in Weaknesses and Questions, these may be limitations.

---

> ### Author Rebuttal · Authors · 2024-08-06
>
> Thank you for your acknowledgements in our method and writing. In the following, we respond to the comments with respect to the weaknesses and questions in turn. All experiments presented in the rebuttal are based on the FAMOUS dataset.
>
> ##### **(1) Concern about frequency feature consistency.**
>
> > There seems to be no guarantee that the scale of the displacement of the query point at each frequency varies with the frequency of interest. Is there some possible constraint on the magnitude of the slope that would be consistent with the frequency of interest?
>
> **RESPONSE**: Yes, your understanding is correct. There is no guarantee for that. But one thing we observed during experiments is that the movement is getting smaller and smaller for the last several pulling since the pulled queries are approaching to the zero-level set. Thus, we use high frequency features as conditions to make the network more sensitive, so that we can capturer high frequency geometry details. We extract different levels of frequency features for the query points in the FFT module and guide multi-step movements in the MSP. As the reviewer mentioned, we separate the frequency features at different scales, so our constraint on frequency feature consistency mainly lies in the direction consistency during multi-step movements. Throughout the continuous displacement process, we use $\mathcal{L}_{grad}$ to constrain the movement direction of the query points at different steps, ensuring consistency across different frequency features.
>
> ##### **(2) Design of $\mathcal{L}_{grad}$.**
>
> > For the purpose of consistency, would it not be appropriate to use the average or correlation rather than argmin to select one of $Q_1$, $Q_2$, or $Q_3$?
>
> **RESPONSE**: Ideally, the movement directions of query points at different steps ($Q_1$, $Q_2$, and $Q_3$) should keep consistent. We ensure this consistency by calculating the gradient similarity loss $L_{grad}$. To prevent significant deviations in the moving direction during training, we use argmin as a constraint, making the network more sensitive to changes in gradient direction. To further illustrate this, we use argmin as the baseline and compare it with using the average in $\mathcal{L}_{grad}$. As shown in Tab. A, when using the average to calculate the similarity of query points at different steps, the CD error slightly increases. Conversely, argmin achieves a similar level of similarity but leads to better performance in terms of the CD metrics.
>
> ##### **(3) Suggestions of Introduction.**
>
> > In the introduction, it would be desirable to specify whether the proposed method is a test-only optimization. Figure 3 in [1] clearly shows the position of the proposed method in relation to related research. My understanding is that the paper plots to the same point as NGLOD, and it is preferable to explicitly state that only test time is considered.
>
> **RESPONSE**: As the reviewer mentioned, our method can learn signed distance functions without introducing extra priors, which is an overfitting based method. In our revision, we will further clarify the differences between the various types of deep learning networks mentioned in the introduction. Additionally, we will provide clear classification explanations based on the references you provided.
>
> ##### **(4) Default setting of Step.**
>
> > After considering both performance metrics and time efficiency, we have set Step=3 by default.", can the authors discuss this trade-off?
>
> **RESPONSE**: We set Step to 3 by default for two reasons: (1) MSP can bring partial accuracy improvement when learning more steps, but more network parameters are also needed. We present comparisons under different steps in Tab. B. (2) Although deeper frequency features can represent more comprehensive information, combining frequency features from different layers can achieve better results. We denote the combination of the frequency features as ${L_1, ..., L_n}$. We conduct the experiment according to the default settings, and the results are shown in Tab. C.
>
> ##### **(5) Related works.**
>
> > "However, inferring implicit functions without 3D supervision requires a very long convergence process, which limits the performance of unsupervised methods in large-scale point cloud data." Can the author provide a specific reference in the paper?
>
> **RESPONSE**: Learning accurate implicit functions with an unsupervised strategy is challenging. Here, we provide some related methods as references: CAP-UDF [2], NP[3], and PCP [4].
>
> ##### **(6) Construction of query points.**
>
> > "To this end, we constrain query points to be as close to their nearest neighbor on S." I understood sampling from the nearest neighbor, but did not understand how Q is constructed.
>
> **RESPONSE**: We construct query points based on the sampling methods of NP. For the input point cloud $s_i \in S$, we sample 80 query points around $s_i$ based on the Gaussian distribution $\mathcal{N}(s_i, \sigma^2)$. We use $\sigma^2$ to control the distance of the query points, we follow NP to set $\sigma^2$ as the square distance between $s_i$ and its 50-th nearest neighbor. We set the default distance as $0.5 \sigma^2$.
>
> [1] Francis Williams, Zan Gojcic, Sameh Khamis, Denis Zorin, Joan Bruna, Sanja Fidler, Or Litany. Neural Fields As Learnable Kernels for 3D Reconstruction. CVPR2022.
>
> [2] Zhou J, Ma B, Li S, et al. CAP-UDF: Learning Unsigned Distance Functions Progressively from Raw Point Clouds with Consistency-Aware Field Optimization[J]. IEEE Transactions on Pattern Analysis and Machine Intelligence, 2024.
>
> [3] Ma B, Han Z, Liu Y S, et al. Neural-pull: Learning signed distance functions from point clouds by learning to pull space onto surfaces[J]. International Conference on Machine Learning, 2021.
>
> [4] Ma B, Liu Y S, Zwicker M, et al. Surface reconstruction from point clouds by learning predictive context priors[C]//Proceedings of the IEEE/CVF Conference on Computer Vision and Pattern Recognition. 2022: 6326-6337.

---

> > ### Comment · Reviewer_NaoM · 2024-08-12
> >
> > I appreciate the polite point-by-point responses from the authors. In particular, I believe that the validity of the paper has been increased by additional results for (2) and (4).
> >
> > I have no further questions at this time.

---

### Official Review · Reviewer_uTM5 · 2024-07-17

**Soundness:** 3
**Presentation:** 3
**Contribution:** 2
**Rating:** 3
**Confidence:** 5

**Summary:**

This paper proposes to learn multi-scale implicit fields from 3D point clouds for accurate optimization of SDFs in a coarse-to-fine manner. The spatial query points are first mapped to frequency features through the FFT module. Then the MSP module is designed to exploit the multi-level frequency features to progressively pull the query points towards to target surface. Various experiments and ablation studies demonstrate the effectiveness of the proposed learning approach.

**Strengths:**

1. The adopted frequency-domain learning paradigm and the pulling-based SDF learning paradigm are effective and advantageous.
2. The experimental comparison shows the performance superiority of the constructed learning pipeline.

**Weaknesses:**

The major weakness of this work is its limited novelty. In general, I think the proposed techniques are just an incremental combination of the pulling-based SDF learning approach [18] (Neural-Pull) and the multiplicative fourier learning approach [14] (MFLOD), although with some marginal modifications in order to combine them. The authors seem to have intentionally overlooked the introduction and detailed discussion of these two types of highly related works. Hence, reviewers that are not very familiar with the two lines of works may overestimate the value and technical novelty of the proposed method. Note that I am not saying that combining some existing techniques is not meaningful. What I mean is that, for this paper, the contribution cannot reach the bar of NeurIPS.

**Questions:**

In lines 82-83, the authors wrote “The creation of 3D shapes LOD usually depends on mesh decimation, which has difficulty in blending between LODs.” I notice that this particular sentence is directly copied from the paper of MFLOD [14]. Explanations are needed.

**Limitations:**

The presented limitations are vague and less constructive.

---

> ### Author Rebuttal · Authors · 2024-08-06
>
> Thank you for your acknowledgements in our method. In the following, we respond to the comments with respect to the weaknesses and questions. All experiments presented in the rebuttal are based on the FAMOUS dataset.
>
> ##### **(1) Novelty.**
>
> > In general, the proposed techniques are just an incremental combination of the pulling-based SDF learning approach [18] (Neural-Pull) and the multiplicative fourier learning approach [14] (MFLOD), although with some marginal modifications in order to combine them. and particular sentence is from the paper of MFLOD.
>
> Although we use feature extraction in MFLOD and pulling operation in NeuralPull as basis, simply combining these two operations does not work well due to unstable optimization or poor performance. Hence, our key contributions include the initialization scheme and multi-step pulling (MSP) modules with multiple constraints, rather than a simple combination. First, our initialization scheme can produce more discriminative frequency features and make each Fourier layer more stable and robust under different frequencies, adapting to more complex geometries, while avoiding gradient vanishing. Second, our MSP module with multiple constraints can infer more accurate SDFs and more consistent gradients using frequency based features as conditions. Whereas NeuralPull infers the SDFs by one-step pulling, it is difficult to directly and accurately reach the target in one step for queries, espcially for the queries that are far from the surface. In addition, NeuralPull does not consider gradient consistency on different level sets, which drastically decreases the accuracy of SDFs. In contrast, MultiPull infers the SDFs by multi-step pulling, the distance-aware constraint pays more attention to queries far from the surface, together with the help of the surface constraint, allowing us to pull any queries to correct locations on the zero-level set. We will make this more clear in our revision.
>
> ##### **(2) Missing Citation.**
> We were supposed to add a citation for the sentence that we get from MFLOD, but we forgot to do that. We will add a citation to this sentence in our revision.

---

> > ### Author Response · Authors · 2024-08-13
> >
> > Dear reviewer uTM5,
> >
> > As the reviewer-author discussion period is about to end, can you please let us know if our rebuttal addressed your concerns? If it is not the case, we are looking forward to taking the last minute to make further explanation and clarification.
> >
> > Thanks,
> >
> > The authors

---

### Author Rebuttal · Authors · 2024-08-06

We thank all the reviewers for their thoughtful comments and valuable feedback. We sincerely thank reviewers uTM5 and NaoM for their supplementary requirements on the paper details and our contribution experiments. Through these experiments, we further demonstrate the scalability and unique contributions of our method. We also appreciate the additional requirements from ZVf1 and VUUa for the experimental content, as well as the analysis of exploratory experiments. These additional experiments significantly enhance the quality and readability of the manuscript . **We provide detailed results in rebuttal and PDF.**

We also value the reviewers' recognition of our paper's writing, presentation, and metrics. We will respond each of the reviewers' questions individually. The additional results and discussions will be added in the revised draft or supplementary materials.

---

### Author Response · Authors · 2024-08-08
**We are happy to take any questions**

Dear reviewers,

We appreciate your comments and expertise. Please let us know if there is anything we can clarify further. We would be happy to take this opportunity to discuss with you.

Thanks,

The authors

---

> ### Author Response · Authors · 2024-08-11
>
> Dear reviewers,
>
> As the reviewer-author discussion period is about to end, we are looking forward to your feedback on our rebuttal. Please let us know if our responses address your concerns. We would be glad to make any further explanation and clarification.
>
> Thanks,
>
> The authors

---

### Decision · Program_Chairs · 2024-09-25

**Decision:**

Accept (poster)

**Comment:**

The paper proposes MultiPull, a multi-scale / multi-step pulling approach to reconstruct a mesh surface from point cloud.  MultiPull trains a neural network to learn a SDF by using multi-scale frequency features. Experiments on several datasets shows that compared to baseline methods, the proposed method has higher reconstruction accuracy and lower errors.

Reviewers are split in their ratings on this work, with three indicating acceptance (ZVf1,VUUa,NaoM) and one voting for reject (uTM5).

The main criticism from reviewer uTM5 is that the proposed method is a combination of pulling-based SDF Neural-Pull [18] and the multiplicative Fourier level of detail MFLOD [14].  In their response, the authors note that it was not trivial to combine the two approaches, as to make use of features from MFLOD and pulling operation from NeuralPull, details of how to initialize and the design of multi-step pulling module was required.  The AC also notes that novel combination of techniques (whether well-known or not) can be valuable (as also noted in the NeurIPS 2024 reviewer guidelines).  As most reviewers find the work to be interesting and there does not appear to be significant flaws with the evaluation, the AC believe this work would be of interest to the NeurIPS community and recommends acceptance.

The authors are encouraged to follow reviewer suggestions to improve the paper:
- Include additional experiments and ablations provided during rebuttal
- Improvements to writing (see detailed comments from VUUa) including adding recent relevant work, discussion of design choices, improve clarity of exposition.
- Please proofread for typos and grammar.  Some (non-exhaustive) examples (in addition to the suggestions from VUUa) are given below:
   - Reword copied sentence (L82-83)
   - Equations
      - Improve notation (see R-ZVf1 comments)
      - L138: Use "\cdot" not "\times"
      - Improve typesetting (use `\sin', '\cos' for sin/cos, '\text{pull}', '\text{recon}', etc for words in equations)
   - Correct number in last row of Table 7 (ZVf1)
   - Line 2: "Latest" => "the latest" (or just "Recent")
   - Line 4: "distnaces" => "distances"
   - Figure 1: "Setp 2" => "Step 2"
   - Table 1: "ShapNet" => "ShapeNet"
   - Line 211: "compare" => "compared"